# Wild Red Deer (*Cervus elaphus*) Do Not Play a Role as Vectors or Reservoirs of SARS-CoV-2 in North-Eastern Poland

**DOI:** 10.3390/v14102290

**Published:** 2022-10-18

**Authors:** Martyna Krupińska, Jakub Borkowski, Aleksander Goll, Joanna Nowicka, Karolina Baranowicz, Vincent Bourret, Tomas Strandin, Sanna Mäki, Ravi Kant, Tarja Sironen, Maciej Grzybek

**Affiliations:** 1Department of Tropical Parasitology, Institute of Maritime and Tropical Medicine, Medical University of Gdansk, 81-519 Gdynia, Poland; 2Department of Forestry and Forest Ecology, University of Warmia and Mazury, 10-727 Olsztyn, Poland; 3Department of Virology, Medicum, University of Helsinki, 00290 Helsinki, Finland; 4Department of Basic Veterinary Sciences, Faculty of Veterinary Medicine, University of Helsinki, 00790 Helsinki, Finland; 5INRAE-Université de Toulouse UR 0035 CEFS, 31326 Castanet Tolosan, France

**Keywords:** deer, SARS-CoV-2, cervid, susceptibility, transmission, spillover

## Abstract

Several studies reported a high prevalence of SARS-CoV-2 among white-tailed deer in North America. Monitoring cervids in all regions to better understand SARS-CoV-2 infection and circulation in other deer populations has been urged. To evaluate deer exposure and/or infection to/by SARS-CoV-2 in Poland, we sampled 90 red deer shot by hunters in five hunting districts in north-eastern Poland. Serum and nasopharyngeal swabs were collected, and then an immunofluorescent assay (IFA) to detect anti-SARS-CoV-2 antibodies was performed as well as real-time PCR with reverse transcription for direct virus detection. No positive samples were detected. There is no evidence of spillover of SARS-CoV-2 from the human to deer population in Poland.

## 1. Introduction

Severe acute respiratory syndrome coronavirus 2 (SARS-CoV-2), the virus responsible for the COVID-19 pandemic, can infect multiple wild and domestic animals [1,2,3,4,5,6]. Due to the possible maintenance, dissemination and evolution of the virus, there is an urgent need to identify and monitor species susceptible to SARS-CoV-2. Animal reservoirs constitute a threat by contributing to animal–animal and animal–human transmission of viruses. Such transmissions can lead to reverse zoonosis (spillback) of novel animal-adapted variants that the human immune system has not encountered before [7,8]. This phenomenon is well documented for SARS-CoV-2 infections of farmed mink (*Neovison vison*). After the first detection of the virus in mink in the Netherlands [4], extensive surveillance of mink farms has been implemented, and SARS-CoV-2-infected mink have been detected worldwide [9,10]. Additionally, in the Netherlands, Denmark and Poland, SARS-CoV-2 strains with an animal sequence signature were detected among farm employees and other contact individuals, indicating spillback from mink to humans [11,12,13,14]. Likewise, the circulation of SARS-CoV-2 among free-roaming wild animals poses challenges for surveillance and control, and abundant animals living close to urban settlements should be paid the utmost attention [8].

In July 2021, antibodies against SARS-CoV-2 were detected among 40% (152/624 samples) of studied white-tailed deer *(Odocoileus virginianus*) during wildlife disease surveillance operations in the United States [15]. Previous studies had shown that white-tailed deer fawns, when experimentally inoculated with SARS-CoV-2, shed infective virus up to 5 days after infection and developed antibodies against the virus [16]. Vertical and horizontal virus transmission has also been documented [17]. In September 2021, Hale et al. employed the real-time PCR method and detected SARS-CoV-2 in 35.8% of 360 free-range white-tailed deer [18]. Moreover, white-tailed deer have been reported to have been exposed to different SARS-CoV-2 variants during separate events [18,19]. Further serological and molecular studies have also investigated the role of white-tailed deer as a reservoir of SARS-CoV-2 [20,21,22,23], and the first suspected deer-to-human transmission has been reported recently [24].

In response to the high numbers of infected deer in North America, the World Organisation for Animal Health (WOAH) recommended monitoring cervids in all regions to better understand SARS-CoV-2 infection and circulation in other deer populations [25]. Recent European efforts have sought to determine whether other deer species can also act as reservoirs for the SARS-CoV-2 virus. Serological studies conducted in Germany, Austria and the United Kingdom found no evidence of spillover of SARS-CoV-2 to cervid species in these regions [26,27,28].

The red deer (*Cervus elaphus*) is one of the most widespread large mammals in Europe [29]. According to the Central Statistical Office in Poland, the national population of red deer increased significantly from 180 thousand to 281.9 thousand over the last eleven years, increasing the risk of exposure to deer in urban environments [30]. Furthermore, an in silico study of ACE2 (angiotensin-I-converting enzyme 2) receptors demonstrated that ACE2 from all three studied Cervidae species—the white-tailed deer, reindeer (*Rangifer tarandus*) and Père David’s deer (*Elaphurus davidianus*)—are at high risk of binding SARS-CoV-2 receptor-binding domain [29]. Apart from K-N substitution, no differences have been found between the red deer and white-tailed deer ACE2 amino acid sequence [28].

As a means for proactive, targeted monitoring of wildlife, we investigated whether a spillover of SARS-CoV-2 occurred among red deer in Poland.

## 2. Materials and Methods

### 2.1. Material Collection and RNA Isolation

Samples were collected from 90 red deer (*Cervus elaphus*) individuals shot by hunters in Warminsko-Mazurskie Voivodeship in north-eastern Poland (Figure 1).

Samples were collected in Strzalowo, Olsztyn, Milomlun, Nidzica and Nowe Ramuki. Table 1 shows the number and status of sampled individuals.

Blood or blood clots were collected and centrifuged to obtain serum. Since animal carcasses were stiff, we used a sterile plastic speculum to open the nasal cavity; then, using a thick swab, we collected nasopharyngeal swabs and preserved them in a virus deactivation buffer at +4 °C. A total of 150 μL from each sample of a swab in inactivation buffer was added to 300 μL of RLT lysis buffer (RNeasy Mini kit, Qiagen, Germany). Samples were mixed by vortexing and incubated for 10 min at room temperature. After incubation, 400 μL of 70% ethanol was added to each sample and mixed by pipetting. The lysate was transferred to a RNeasy Mini spin column with a collection tube and centrifuged for 1 min at 13,000 RPM. Columns were washed once with 700 μL RW1 and twice with 500 μL RPE. Between every wash, the columns were centrifuged and the flow-through was discarded. Elution was performed by adding 50 μL of PCR-grade water to the column and incubating for 2 min. Columns were placed into new tubes and centrifuged at 13,000 RPM for 1 min. After isolation, the samples were processed. We tested 90 nasopharyngeal swab samples by RT-rtPCR and 90 serum samples by immunofluorescent assay (IFA). No human-origin samples were processed at the same time in the laboratory.

### 2.2. SARS-CoV-2 Case Definition

A SARS-CoV-2-positive individual was defined as suggested by WOAH [31]. Deer are considered SARS-CoV-2 positive if: SARS-CoV-2 has been isolated from a sample taken directly from an animal (nasal swab, oropharyngeal swab) or viral nucleic acid has been identified in a sample taken directly from an animal, giving cause for the suspicion of a previous association or contact with SARS-CoV-2 by (a) targeting at least two specific genomic regions at a level indicating the presence of infectious virus or (b) targeting a single genomic region followed by sequencing of a secondary target.

### 2.3. Real-Time RT-PCR 

For each sample, the reaction mixture was prepared using a TaqPath™ 1-Step RT-qPCR Master Mix (ThermoFisher Scientific, Waltham, MA, USA), polymerase, DEPC-treated water (EURx, Gdańsk, Poland), primers and probes for the RNA-dependent RNA polymerase (RdRp) and envelope (E) genes [31] in white 8-well qPCR strips with optical clear caps. Positive control plasmids were prepared in-house with the RdRp and E genes and a no-template control (NTC) containing DEPC-treated water instead of template reactions. Reactions were mixed and loaded into a Light Cycler 480 (Applied Biosystems, Waltham, USA). Cycling conditions were Uracil N-glycosylase (UNG) incubation for 2 min at 25 °C, RT incubation for 15 min at 50 °C, and enzyme activation for 2 minutes at 95 °C, followed by 40 amplification cycles consisting of 3 seconds at 95 °C, and 30 s at 60 °C. After each amplification cycle, the signal from each sample was measured in both the FAM (RdRp gene) and HEX (E gene) channels. Samples with Cp < 35 for either gene were considered positive for SARS-CoV-2.

### 2.4. Immunofluorescent Assay (IFA) for Antibodies against SARS-CoV-2 Detection

Deer serum samples were analysed using an immunofluorescence assay (IFA) with seropositive human serum as a positive control as previously described [3,32,33,34]. Serum samples were diluted 1:10 in PBS, and the reactivity of the samples to SARS-CoV-2 was tested with SARS-CoV-2-IFA. Infected Vero E6 cells were detached with trypsin, mixed with uninfected Vero E6 cells (in a ratio of 1:3), washed with PBS, spotted on IFA slides, air-dried, and fixed with acetone. The slides were stored at –70 ℃ until use. We used rabbit anti-deer IgG fluoresceinisothiocyanate labeled as a conjugate (LGC Sera Care, Milford, CT, USA). The slides were read under a fluorescence microscope, and pictures were taken with a ZOETM fluorescent cell imager (BioRad, Hercules, USA).

## 3. Results

Real-time RT-PCR and IFA approach failed to detect any SARS-CoV-2 positive samples among the 90 assayed. The estimated prevalence and seroprevalence for SARS-CoV-2 in the investigated red deer population was 0% [0.0–6.7].

## 4. Discussion

We did not detect either the SARS-CoV-2 virus or antibodies against the virus in wild red deer in Poland. There is no evidence that spillover of SARS-CoV-2 from human to deer populations occurred in the studied territory. Our results align with other European countries’ reports on wild deer populations [26,27,28]. Comprehensive studies from the United Kingdom [26], Austria and Germany [28] showed no sign of SARS-CoV-2 in several deer species. This result supports the hypothesis that wild deer are not currently a reservoir for SARS-CoV-2 in Europe.

In contrast to European results, the high prevalence and seroprevalence of SARS-CoV-2 in white-tailed deer in North America have been reported in several studies [15,18]. Multiple spillovers of SARS-CoV-2 from humans to white-tailed deer have been documented, along with deer-to-deer transmission. A recent report by Pickering et al. [24] found an epidemiologically linked human case indicating spillback from deer to humans.

There are several factors to consider to explain observed differences in infection rates in deer populations from North America and Europe. The high prevalence in white-tailed deer may be explained by its ACE2 receptor specificity. In silico modelling studies suggest that other deer species might also be susceptible to SARS-CoV-2 [28,35]. Although K-N substitution between red deer and white-tailed deer ACE2 sequence occurs, it fails to explain the lack of detected infections in other European deer species with an ACE2 receptor sequence identical to white-tailed deer. Crucial factors to consider are the deer population distribution, ecology, and behavioural differences. White-tailed deer are often reported to inhabit urban and peri-urban environments [36,37,38].

In contrast, red deer found in European forests are rarely reported to visit human settlements [39]. They are considered to be timid and avoid contact with humans. This may explain our and other groups’ results, where red deer were found to be negative for SARS-CoV-2 screening [26,27,28]. Although routes of white-tailed deer infection are unclear, deer social behaviour might be the factor facilitating the quick spread of the pathogen. Female white-tailed deer live in small herds, while males have broader territories and social contacts, increasing the risk of contact with the virus [40]. Intermediate hosts, such as mink, enabling transmission also cannot be ruled out [24]. Differences in hunting practices should also be considered; for example, animal baiting is allowed in over 20 out of 50 states in North America, which may cause indirect contact with humans, while this practice is forbidden in Poland, as in many other European countries [28].

This study suggests that common wild European red deer are not currently supporting SARS-CoV-2 infections. However, considering reports from North America, it is necessary to monitor wildlife for spillover and spillbacks. Biomonitoring is one of the most effective methods of predicting and preventing possible epidemics [33,41,42,43,44]. Searching for novel hosts or reservoirs of zoonotic pathogens should be a priority for public health and wildlife management institutions [32,45]. The introduction of SARS-CoV-2 to wildlife has caused the establishment of animal reservoirs (i.e., white-tail deer). Therefore, all efforts should be made to reduce the risk of new variant emergence and to protect both humans and wildlife (FAO, WHO, WOAH, 2021 [46]). We believe that preparation for a “Pathogen X” pandemic should employ a “One Health” approach with a strong emphasis on monitoring both domestic and wild animals.

## Figures and Tables

**Figure 1 viruses-14-02290-f001:**
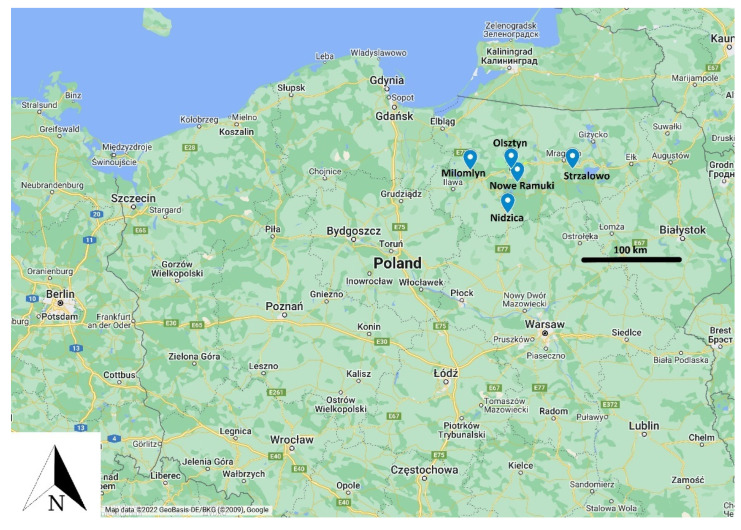
Four hunting districts are located in north-eastern Poland. The black bar indicates 100 km distance (Map data, Google, 2022).

**Table 1 viruses-14-02290-t001:** Location, status, and number of red deer (*Cervus elaphus)* individuals sampled.

Hunting District/Deer	Doe	Bull	Fawn	Total
Strzalowo	16	2	0	18
Olsztyn	12	4	3	19
Milomlyn	4	0	1	5
Nidzica	8	4	4	16
Nowe Ramuki	20	7	5	32
Total	60	17	13	90

## Data Availability

Data are available from authors upon request.

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
