# Peer review of "Wild Red Deer (Cervus elaphus) Do Not Play a Role as Vectors or Reservoirs of SARS-CoV-2 in North-Eastern Poland"

_viruses, 2022, doi:10.3390/v14102290_

Round 1

Reviewer 1 Report

I have no particular comments. I ask the authors, however, to review the references section well

Author Response

We thank Reviewer 1 for his time to read and assess our work. We checked and amended the reference section according to the journal requirements. 

Reviewer 2 Report

The article "Wild red deer (Cervus elaphus) do not play a role as vectors or reservoirs of SARS-CoV-2 in north-eastern Poland." is a valuable contribution for the understanding of SARS-CoV-2 epidemiology in wildlife. The publication of negative results is necessary to expand our knowledge in this field.

Minor comments: 

ABSTRACT:

- Please replace: "to evaluate deer exposure to SARS-CoV-2 in Poland, we sampled 90 reed deer individuals shot by hunters in five hunting districts in north-eastern Poland." by "to evaluate deer exposure and/or infection to/by SARS-CoV-2 in Poland, we sampled 90 reed deer individuals shot by hunters in five hunting districts in north-eastern Poland.  

- Provide the overall number of individuals tested in the sentence: "In July 2021, antibodies against SARS-CoV-2 were detected among 40%..."

- Provide the reference number to the sentence: In September 2021, Hale et al. employed...

- Please provide an adequate caption for Table 1, including the common and scientific name of red deer. Do you have information about the sex of the tested fawns?  Please provide a caption for Figure 1. 

- Please specify in your results that you tested 90 nasopharyngeal swab samples by RT-rtPCR and 90 serum samples by IFA.

- Please provide  adequate references to the sentence: "Comprehensive studies from the United Kingdom, Austria and Germany..."

Author Response

The article "Wild red deer (Cervus elaphus) do not play a role as vectors or reservoirs of SARS-CoV-2 in north-eastern Poland." is a valuable contribution for the understanding of SARS-CoV-2 epidemiology in wildlife. The publication of negative results is necessary to expand our knowledge in this field.

*OUR RESPONSE: We thank Reviewer 2 for his time and effort. We are more than glad to hear that publishing negative results is important and leads to filling the gap in the SARS-CoV-2 virus spread within animal species.

Minor comments: 

ABSTRACT:

- Please replace: "to evaluate deer exposure to SARS-CoV-2 in Poland, we sampled 90 reed deer individuals shot by hunters in five hunting districts in north-eastern Poland." by "to evaluate deer exposure and/or infection to/by SARS-CoV-2 in Poland, we sampled 90 reed deer individuals shot by hunters in five hunting districts in north-eastern Poland.  

 *OUR RESPONSE: Abstract was amended as suggested above.

- Provide the overall number of individuals tested in the sentence: "In July 2021, antibodies against SARS-CoV-2 were detected among 40%..."

 *OUR RESPONSE:  We amended the sentence to read: In July 2021, antibodies against SARS-CoV-2 were detected among 40% (152/624 samples) of studied white-tailed deer (Odocoileus virginianus) during wildlife disease sur-veillance operations in the United States [13].

- Provide the reference number to the sentence: In September 2021, Hale et al. employed...

*OUR RESPONSE:  Missing reference was added.

- Please provide an adequate caption for Table 1, including the common and scientific name of red deer. Do you have information about the sex of the tested fawns?  Please provide a caption for Figure 1. 

*OUR RESPONSE: We provided a caption for Table 1 -> “Table 1. Location, status and the number of red deer (Cervus elaphus) individual sampled” as suggested. No data on sex in fawns was collected during the sampling.

- Please specify in your results that you tested 90 nasopharyngeal swab samples by RT-rtPCR and 90 serum samples by IFA.

*OUR RESPONSE: We provided this information in Material and Methods section.

- Please provide  adequate references to the sentence: "Comprehensive studies from the United Kingdom, Austria and Germany...".

*OUR RESPONSCE: We provided missing references